# Comparison of the Ammonia Trapping Performance of Different Gas-Permeable Tubular Membrane System Configurations

**DOI:** 10.3390/membranes12111104

**Published:** 2022-11-05

**Authors:** María Soto-Herranz, Mercedes Sánchez-Báscones, María Cruz García-González, Pablo Martín-Ramos

**Affiliations:** 1Department of Agroforestry Sciences, ETSIIAA, University of Valladolid, Avenida de Madrid 44, 34004 Palencia, Spain; 2Instituto Universitario de Investigación en Ciencias Ambientales de Aragón (IUCA), EPS, Universidad de Zaragoza, Carretera de Cuarte s/n, 22071 Huesca, Spain

**Keywords:** ammonia recovery, gas-permeable membrane, submerged GPM system, suspended GPM system

## Abstract

The technology of gas-permeable tubular membranes (GPMs) is promising in reducing ammonia emissions from livestock manure, capturing NH_3_ in an acidic solution, and obtaining final products suitable for valorization as fertilizers, in line with the principles of the circular economy. This study aimed to evaluate the performance of several e-PTFE membrane systems with different configurations for the recovery of NH_3_ released from pig slurry. Ten different configurations were tested: only a submerged membrane, only a suspended membrane in the same chamber, only a suspended membrane in an annex chamber, a submerged membrane + a suspended membrane in the same chamber, and a submerged membrane + a suspended membrane in an annex chamber, considering in each case the scenarios without and with agitation and aeration of the slurry. In all tests, sulfuric acid (1N H_2_SO_4_) was used as the NH_3_ capture solution, which circulated at a flow rate of 2.1 L·h^−1^. The results showed that NH_3_-N removal rates ranged from 36–39% (for systems with a single submerged or suspended membrane without agitation or aeration of the slurry) to 70–72% for submerged + suspended GPM systems with agitation and aeration. In turn, NH_3_-N recovery rates were found to be between 44–54% (for systems with a single membrane suspended in an annex compartment) and 88–91% (for systems based on a single submerged membrane). However, when choosing a system for farm deployment, it is essential to consider not only the capture and recovery performance of the system, but also the investment and operating costs (ranging from 9.8 to 21.2 €/kg N recovered depending on the selected configuration). The overall assessment suggests that the simplest systems, based on a single membrane, may be the most recommendable.

## 1. Introduction

Ammonia (NH_3_) emissions from livestock farming are a major source of odor and environmental pollution [1,2]. In addition, they lead to a significant loss of N, a valuable plant nutrient. Therefore, there is great interest in the application of control technologies to reduce NH_3_ emissions through N capture and recovery, which would partially offset the implementation and operation costs associated with such control technologies through the revenue obtained from the sale of the fertilizer product [3,4].

Among the technologies used to remove NH_3_ generated in livestock houses are the treatment of the exhaust air from the house using scrubbing or filtration techniques (biotrickling or biofilters) and chemical amendments [5,6,7]. However, in these cases, either the retention of N involves high costs, or, even if the N is retained without volatilizing in the litter, the NH_3_ is not recovered as a separate product, as is the case with scrubbing techniques.

This study investigates the use of an alternative technology based on expanded polytetrafluoroethylene (e-PTFE) gas-permeable passive membranes [3,8,9] that allows capturing NH_3_ (g) in an acidic solution and recovering N in a concentrated and purified form, not dependent on intense air movement. This membrane material was selected as it is microporous, flexible, and hydrophobic, with a high permeability rate for low-pressure gas flow differentials between the inside and outside of the tube. In previous studies, e-PTFE membranes have led to high NH_3_ recoveries from different sources, such as chicken manure, pig manure, and anaerobically digested slurry [10,11,12], effectively reducing the TAN concentration in the sources. Further, it has also demonstrated high NH_3_ recovery efficiencies from livestock housing air [3,13,14].

The performance of this technology depends on the availability of NH_3_ in the TAN source, where NH_3_ and NH_4_^+^ are in equilibrium, which in turn depends on factors such as the pH and temperature of the TAN source [3,15]. The concentration of NH_3_ in the emitting source, the surface area of the membrane, the composition of the capture solution used, and its rate of circulation also influence the efficiency of the process [9,15,16,17,18,19,20].

To date, these membranes have generally been used in the submerged configuration for the capture of N from slurry and digestate [16,21]. More recently, the use of suspended membrane configurations has also been proposed [3,13,14], and some authors have explored the possibility of combining both systems [9]. Submerged GPM configurations yield higher NH_3_ capture efficiencies than suspended GPM systems, as they are in direct contact with the N-emitting source. However, agitation and aeration of the slurry to achieve these efficiencies involve a significant discharge of NH_3_ (g) into the medium, and the submerged GPM system is unable to capture all the N from the source before the discharge of NH_3_ occurs. To improve the NH_3_ capture performance, the possibility of combining both GPM systems (submerged and suspended) has been considered.

Nevertheless, it is difficult to compare the efficiency of the different configurations due to the diversity of working conditions in the different studies (very different N concentrations in the emitting solution, different membrane characteristics and surfaces, variety of capture solutions and their pumping speeds, etc.).

The aim of this research was to carry out a comparative study in systematized conditions that would allow evaluating the response of different gas-permeable membrane systems on the NH_3_ diffusion flux and the NH_3_ extraction efficiency from pig slurry in a reliable way, to select the most advantageous ones with a view to their application on an industrial scale (for NH_3_ capture from livestock buildings, in manure storage facilities, in composters, etc.).

## 2. Materials and Methods

### 2.1. Origin of the Slurry

Pig slurry was collected from a slurry pond of a mother sow farm located in Dueñas (Palencia, Spain) and was transported to the laboratory and kept at 4 °C until the experiments were carried out.

### 2.2. Experimental Design

The schematics of the different GPM membrane systems used in these experiments for NH_3_ recovery are shown in Appendix A.

The assembly consisted of a closed methacrylate chamber with a capacity of 30 L, inside which 2 L of pig slurry were placed together with different membrane systems, installed inside or in a 15 L annex compartment. The evaluated systems were the following:S1: Submerged GPM without slurry agitation or aeration.S2: Suspended GPM installed in the headspace of the treatment chamber without slurry agitation and aeration.S3: Suspended GPM installed in the headspace of a chamber attached to the treatment chamber where the slurry was without agitation or aeration.S4: Submerged and suspended GPMs in the same treatment chamber without agitation or aeration of the slurry.S5: Submerged GPM without slurry agitation or aeration and suspended GPM installed in the headspace of a chamber attached to the treatment chamber where the slurry was located.S6: GPM submerged with slurry agitation and aeration of 0.24 L air/(L slurry·min).S7: Suspended GPM installed in the headspace of the treatment chamber with slurry agitation and aeration of 0.24 L air/(L slurry·min).S8: Suspended GPM installed in the headspace of a chamber attached to the treatment chamber where the slurry was located with agitation and aeration of 0.24 L air/(L slurry·min).S9: Submerged and suspended GPMs in the same treatment chamber with slurry agitation and aeration of 0.24 L air/(L slurry·min).S10: Submerged GPM with slurry agitation and aeration of 0.24 L air/(L slurry·min) and suspended GPM installed in the headspace of a chamber attached to the treatment chamber where the slurry was located.

All experiments were carried out at the same time in the same laboratory, in controlled conditions. The temperature of the slurry and uptake solutions was monitored using a pH and temperature electrode, giving mean values of 23.4 ± 2.5 and 24.4 ± 2.4 °C, respectively. The pH of the slurry in the chambers was maintained during the experiments at an average value of 8.3 ± 0.2.

In all adjoining chamber systems, the air was drawn in at a flow rate of 200 L·h^−1^ from the treatment chamber to the adjoining compartment, with air recirculation back to the treatment chamber.

The airflow rate selected for slurry aeration was low to effectively maintain pH and avoid NH_4_^+^ nitrification (which lowers slurry pH). This selected aeration rate showed good results in previous studies [22,23]. In the aerated treatments, the air was supplied by an aquarium air pump from the bottom of the experimental chamber through a porous stone, and the supply was controlled by an airflow meter (Instruments Direct, Canton, GA, USA). Given that aeration was combined with agitation, uniformity throughout the volume was assured. Additionally, 10 mg·L^−1^ of a commercial nitrification inhibitor (allylthiourea) was added to the slurry in all experiments.

For NH_3_ capture, an e-PTFE membrane (Zeus Industrial Products Inc., Orangeburg, SC, USA) with an outer diameter of 5.2 mm, a wall thickness of 0.64 mm, a polymer density of 0.95 g·cm^−3^, a porosity < 60%, an average pore length of 12.7 ± 5.9 µm, and an average pore width of 1.3 ± 0.9 µm was used. The pores of the e-PTFE membrane were elongated in the extrusion process during its fabrication [24]. The length of the GPM tubing was 100 cm in all experiments, giving a surface area of 163.4 cm^2^ per installed membrane (i.e., 163.4 cm^2^ for single membrane systems and 326.8 cm^2^ for double membrane systems). The choice of this membrane length/surface was supported by previous studies [19].

Each membrane circuit had an associated 1-L 1N H_2_SO_4_ reservoir for NH_3_ capture. Acid was pumped through the membranes using a peristaltic pump (Pumpdrive 5001, Heidolph, Schwabach, Germany) at a flow rate of 2.1 L·h^−1^.

### 2.3. Analysis Methodology

An orifice in the chambers, which was opened for a few seconds and in a punctual way, was used to carry out the slurry sampling. This also served to balance the pressure between the inside and outside of the chamber. Slurry and acidic solution samples were collected in triplicate (25 mL) three times a week.

Analyses of pH, temperature, and NH_3_-N concentration were performed on all samples. pH and temperature were measured with a Crison GLP22 m electrode (Crison Instruments S.A., Barcelona, Spain), and the NH_3_-N concentration was determined by distillation (Kjeltec^TM^ 8100; Foss Iberia S.A., Barcelona, Spain) by collecting the distillate in borate buffer and subsequent titration with 0.2N HCl [25].

### 2.4. Data Calculation

Free ammonia (FA) was quantified according to Equation (1) [26]:(1)$$\left[{\mathrm{NH}}_{3}\right]/\left[{\mathrm{TNH}}_{3}\right]=(1+{\left(\frac{10^{-\mathrm{pH}}}{10^{-\left(0.09018+2729.92/T\right)}}\right)}^{-1},$$
where [NH_3_] is the concentration of free ammonia, [TNH_3_] is the total concentration of ammonia, and T is the temperature (K).

The amount of NH_3_ removed (mg NH_3_-N) was determined as the difference between the amount of NH_3_ in the slurry at the beginning and the end of the experiment. The mass of NH_3_ recovered (mg NH_3_-N) refers to the amount of NH_3_-N captured at the end of the experiment in the acidic solution. The N recovery efficiency (%) was estimated by dividing the recovered mass by the removed mass. The mass flux of NH_3_-N across the membrane (J, expressed in mg NH_3_-N·cm^−2^·d^−1^) was determined considering the mass of N captured per day and the surface area of the GPM pipe.

### 2.5. Statistical Analyses

Data were tested for homogeneity and homoscedasticity using the Shapiro–Wilk and Levene tests, respectively, after which a two-way analysis of variance (ANOVA) was performed, followed by a post hoc comparison of means using Tukey’s test at *p* < 0.05. The statistical software R was used for these statistical analyses [27].

### 2.6. Economic Analysis of the Different GPM Systems

To compare the different GPM systems used to reduce NH_3_ emissions, an economic analysis of each system was performed. Operating costs associated with the electrical energy consumption of the elements used in each case (peristaltic pump, agitator, aeration pump, and suction pump) and costs associated with the price of the membrane and acid were considered. The values used in these calculations were based on the experimental data from this study, along with the following assumptions:The annual production of pig slurry on the farm is 2795 m^3^ per year (i.e., 7.7 m^3^·d^−1^), resulting from 1300 animals producing 2.15 m^3^ slurry·year^−1^ each.Pig slurry contains between 931.6 and 1032.1 mg TAN·L^−1^ for each system evaluated.A TAN removal target for pig slurry of approximately 35% is proposed.TAN recovery efficiencies and TAN recovery rates at the end of the 7-day experimental period have been taken as a reference for each evaluated system.The cost of the membrane is 115 €·m^−2^ (1.88 €·m^−1^) [12] and a 10% replacement per year is considered.Annualized equipment costs are calculated assuming a useful life of 10 years and an interest rate of 8% [12].The amount of H_2_SO_4_ (98%) required to capture TAN ranges from 446.5 to 494.7 kg acid/kg recovered TAN for the systems evaluated in this study.

## 3. Results and Discussion

### 3.1. Nitrogen Removal from Pig Slurry

The concentration of NH_3_-N present in the slurry decreased during the experimental period for all GPM systems (Table 1), with reductions ranging from 36 to 72%.

The most efficient systems in terms of NH_3_-N removal were S8, S9, and S10, all of which are systems with slurry agitation and aeration. In general, agitation and aeration of the N emission source allow the pH of the slurry to be kept high while preventing the nitrification of NH_4_^+^. Aeration causes the release of OH^−^, which raises the pH of the slurry, reducing its alkalinity according to Equation (2):(2)$${\mathrm{HCO}}_{3}^{-}+\;\mathrm{air}\to {\;\mathrm{OH}}^{-}+{\;\mathrm{CO}}_{2},$$

The increase in pH due to aeration, therefore, affects the formation of NH_3_ and the efficiency of the membrane system to recover N from the slurry, acting as indicated in Equation (3):(3)$${\mathrm{NH}}_{4}^{+}+{\mathrm{OH}}^{-}↔{\;\mathrm{NH}}_{3}+H_{2}O,$$

The recovery of NH_3_ (g) through the gas-permeable membrane causes an increase of acidity in the slurry, as H^+^ does not permeate the hydrophobic membrane. Therefore, the recovery of the NH_3_ by the membrane, from a net perspective, causes an acidification of the slurry and a reduction of the alkalinity. Daguerre et al. [28] studied this behavior in further detail, concluding that the release of OH^−^ from natural carbonates increased the manure pH and promoted gaseous ammonia formation and membrane uptake, and, at the same time, the recovery of gaseous ammonia through the membrane acidified the slurry. Similar behavior was observed by other authors, such as Oliveira et al. [29].

Another effect induced by aeration was the increase of FA content in the slurry, which increased in values ranging between 203.2 and 359.4 mg·L^−1^ from the beginning to the end of the experiment in the systems with slurry aeration (due to the increase in slurry pH), while in the systems without aeration it ranged between 42.7 and 315.7 mg·L^−1^. Three cases were observed in which the FA content decreased at the end of the trial in comparison to the beginning, these being systems S3, S6, and S7. According to García-González and Vanotti [30], since the FA content was always above 40 mg·L^−1^ (except for the three cases mentioned), active permeation of NH_3_ through the membranes would be guaranteed, favoring an optimal NH_3_ capture process.

Our results were consistent with those obtained by García-González et al. [22], who determined a 33% higher NH_4_^+^ removal efficiency for a submerged e-PTFE membrane treatment in aerated slurry than in non-aerated slurry during an 18-day experiment. In our case, aeration using a submerged membrane (S6) improved the NH_3_-N removal efficiency by 23% compared to the same system without aeration (S1) in 7 days. In addition, the aforementioned authors indicated that, in the test with submerged membrane without slurry aeration, the FA concentration remained below 100 mg N·L^−1^ during the experiment, compared to FA concentrations of up to 250 mg N·L^−1^ in the treatment with aerated slurry. In this regard, the results of this study differed from those mentioned above: in the submerged membrane trial without slurry aeration (S1), an increase in FA concentration was observed as the experiment progressed, while in the submerged membrane trial with slurry aeration (S6) this concentration decreased over time.

Likewise, the S4 configuration was similar to the one used by Majd and Mukhtar [9], who used suspended and submerged membrane systems at the same time for the capture of NH_3_ from slurry without agitation and aeration. In this study, our results showed a removal of NH_3_-N from the slurry of 66% in 7 days, higher than that found by the authors mentioned above, who determined a removal percentage of NH_3_-N of 48% in 15 days of experimental operation with pH control. This difference can be attributed to the fact that the pig slurry used as a source of N emission in this study presented a notably higher initial NH_3_-N concentration (2170 mg·L^−1^) than the one used in their experiments (117 mg·L^−1^).

The lowest NH_3_-N reductions in the slurry were observed in systems S1 and S2 (39% and 36%, respectively) (Table 1), corresponding to submerged and suspended membrane systems without slurry agitation and aeration, respectively. In both systems, a low amount of NH_3_ was released into the air in the chamber and, therefore, a lower amount was recovered in the acidic solution compared to other systems. For comparison purposes, other authors such as Vanotti and Szogi [11] and García-González and Vanotti [30] obtained NH_4_^+^ removal percentages in treatments with submerged e-PTFE membranes without aeration or agitation of the slurry of 50 and 57% with respect to the initial content of NH_4_^+^ in the slurry in 9 and 20 days, respectively.

The rest of the systems (S3, S5, and S7) showed intermediate NH_3_-N removal values, in the 60–70% interval.

### 3.2. Nitrogen Recovery from Pig Slurry

The concentration of NH_3_-N in the capture solution increased in all cases during the experimental period, exceeding the threshold of 1000 mg N·L^−1^ (Appendix A). The highest NH_3_-N concentration was obtained with systems S9, S10, S5, and S4. All of these systems had two membranes (submerged and suspended), either in the same compartment or in an annexed one, with and without continuous agitation of the slurry, respectively. Such a high concentration would be due to the sum of NH_3_ recoveries in both membranes. If the NH_3_-N recovery per specific membrane surface area is plotted instead (Figure 1), it becomes apparent that these double-membrane systems would not be so efficient, as will be discussed below. Comparing homologous pairs without and with aeration, the highest increase in NH_3_-N recovery per membrane surface area was obtained in the S2–S7 pair (45%), followed by S1–S6 (38%).

Regarding the mass of recovered NH_3_-N and the NH_3_-N recovery efficiency, it is interesting to make a comparison between homologous systems, in pairs: S1–S6, S2–S7, S3–S8, S4–S9, and S5–S10. Regarding the first parameter, significant differences were observed only in the following pairs of counterparts: S1–S6, S2–S7, and S3–S8. In contrast, no significant changes in the mass of NH_3_-N recovered were observed for the S4–S9 and S5–S10 pairs.

In the three cases in which significant differences were observed, they were systems with a single membrane, and the difference between the systems was a result of the presence or absence of agitation and aeration in the slurry. As indicated above, aeration allows the slurry pH to be kept high, which favors the presence of more NH_3_ (g) in the air of the treatment chamber, which explains why a significantly higher recovery was obtained than in the same type of system in which there was no aeration and agitation of the slurry.

As for the recovery efficiency, it should be noted that the differences were minimal, within measurement error (±10%): 3% for S1–S6, 2% for S2–S7, 10% for S3–S8, 8% for S4–S9, and 1% for S5–S10. This suggests that the efficiency of NH_3_-N recovery depends on the system configuration (i.e., a submerged GPM, a suspended GPM, or a suspended GPM in an annex chamber, or two GPM, either in the same chamber or in an annex chamber) and not on the presence/absence of agitation and aeration of the slurry.

Comparing the results obtained with those mentioned in the literature, the recovery efficiency values for systems based on a single submerged GPM (S1–S6), of 88 and 91% (Table 1), were comparable to those reported by Vanotti et al. [31], Oliveira Filho et al. [29], and Riaño et al. [23], who also obtained high ammonia recovery (~90%) using a submerged membrane with low aeration of raw or digested swine manure. However, they were lower than those reported by García-González et al. [22], who, in a process of capture of ammonia from raw swine manure using a submerged gas-permeable membrane, were able to achieve an overall recovery of ammonia of 99% with respect to the initial NH_4_^+^ present in the manure, regardless of the presence or absence of aeration. Dube et al. [12] also achieved 96–98% recoveries of NH_4_^+^ present in digested swine manure in 5 days using a submerged tubular gas membrane, together with low-rate aeration and a nitrogen inhibitor, while recoveries obtained using submerged membrane without digested swine manure aeration were 76–95% over a period of 25 days.

In the S2–S7 systems, based on a single GPM suspended in the air chamber in which the slurry was located, the efficiencies were 69 and 71%, notably lower than that obtained by Soto-Herranz et al. [19] (96%) for a similar suspended e-PTFE system for an acid flow rate of 2.1 L·h^−1^ in an experimental period of 7 days, similar to the one used in this study. The amount of NH_3_-N recovered was also notably lower (1083 vs. 1602 mg N). These differences can be attributed to the fact that in that experiment a synthetic solution was used as a source of N emission instead of pig slurry, with a TAN of 6000 mg NH_3_-N·L^−1^ (vs. 4308 mg NH_3_-N·L^−1^ in this study). On the other hand, the values obtained in this study were significantly higher than those obtained in an NH_3_ capture system that used a non-hydrophobic and non-microporous membrane and a 2:3 (*v*/*v*) ratio of emitting solution (pig slurry) to acidic solution: 46% in 24 h and 59% in 48 h [32].

On the other hand, when comparing the results of S1–S2 (submerged vs. suspended GPM without slurry agitation and aeration) and S6–S7 (submerged vs. suspended GPM with slurry agitation and aeration), significant differences in ammonia capture were observed as a function of membrane position. Such differences can be attributed to the fact that the membranes suspended in the headspace of the chamber are not in direct contact with the emission source and, therefore, the NH_3_ capture is lower than in submerged GPM systems, given that the concentration in the air is lower than that present in the liquid. However, it should be noted that this result does not coincide with that reported by Rothrock et al. [3], who found that the relative position of the tubular membranes (above, directly on, or below the poultry litter surface) did not significantly affect the total mass of NH_3_ recovered by the system or the mass of NH_4_-N remaining in the poultry litter after volatilization. It is also worth mentioning that, in the present work, no reduction in the TAN recovery rate was detected over time in those configurations in which the membrane was submerged, suggesting that fouling—which determines the lifetime of GPMs and affects their economic viability [33]—was not relevant. However, further studies with longer run times would be needed to determine the effect of membrane configuration on the anti-fouling performance.

In S3–S8 systems, in which the membrane was suspended in an annex compartment, forced ventilation toward this compartment (aeration), as previously indicated, should favor the presence of more NH_3_ (g) available to be aspirated and concentrated in the annex chamber. It should be noted that the capture process of NH_3_ in the acidic solution worked better in the S3 system than in S8, which had agitation and aeration of the slurry. This unexpected result could have its origin in the fact that agitation leads to a higher NH_3_ emission, part of which would be lost before being recovered by the membrane, while in the system without agitation only the phenomenon of gas diffusion through the membrane would influence NH_3_ capture. The NH_3_-N recovery efficiencies achieved in these systems, 54 and 44%, cannot be reliably compared with the literature: to the best of the authors’ knowledge, this type of system has only been previously tested by our group, with values of 4108 g NH_3_-N in 232 days in a pig farm and 794 g NH_3_-N in 256 days in a poultry farm under real conditions, not under laboratory ones.

On the other hand, the results obtained for systems S4–S9 and S5–S10 are interesting. In both cases, double membranes were used: in the first pair of systems, both membranes (submerged and suspended) were located inside the same treatment chamber and, in the second pair, the suspended GPM was installed in an annex compartment. Each pair of systems was studied with the presence or absence of agitation and aeration. No significant differences in NH_3_ capture were observed between homologous or non-homologous pairs. This suggests that when using a double membrane system, it would not be necessary to apply aeration and agitation to the slurry to favor the capture of NH_3_-N. Furthermore, concerning the systems in which air was extracted from the treatment chamber to an external compartment in which a suspended GPM membrane was installed (S5–S10), no significant differences were observed in the mass of NH_3_-N recovered in the capture solution between homologous pairs. This is especially interesting, since air extraction into an external compartment would reduce the effect of agitation and aeration on the NH_3_ recovery performance of the systems. The NH_3_-N recovery efficiencies achieved in these systems were 87% and 79% for S4–S9 and 85–86% for S5–S10 (Table 1). Majd and Mukhtar [9] used an experimental design similar to S4 and, although they did not expressly report recovery efficiency data, it is possible to calculate it based on the information provided in their study, obtaining a joint NH_3_-N recovery efficiency of 79%. Therefore, the results obtained in this study would be consistent with those reported by these authors.

If the impact on NH_3_-N recovered from the use of the double membrane system compared to the single membrane system is evaluated, significant differences can be observed between systems without aeration or agitation of the slurry with a single membrane (S1 and S2) and the homologous double membrane system (S4). Such differences are reasonable, given that in the latter system the membrane surface area in contact with NH_3_, either in the slurry or in the air, is twice than in the former two systems. On the other hand, when comparing systems with aeration and slurry agitation with a single membrane (S6 and S7) vs. their double membrane counterpart (S9), differences were only observed between S7 (suspended membrane) and S9 (double membrane), not between S6 (submerged membrane) and S9 (double membrane). This result is interesting, given that it raises the possibility that, in cases in which agitation and aeration are applied to the slurry, it would not be necessary to use a double membrane to attain a high NH_3_-N capture: using only a membrane submerged in the slurry would be enough to obtain good capture yields.

### 3.3. N Flux Rates

In the submerged GPM systems (S1 and S6), without and with slurry aeration and agitation, respectively, the N flux rates were 12.4 and 20.4 g N·m^−2^·d^−1^ (Table 1), higher than those obtained by González-García et al. [34] (8 g N·m^−2^·d^−1^) and García-González and Vanotti [22] (9.5 g N·m^−2^·d^−1^) in submerged systems with low slurry aeration.

N flux rates in suspended GPM systems (S2 and S7), without and with aeration and agitation, respectively, were 9.5 and 16.8 g N·m^−2^·d^−1^, both within the range of values provided by Rothrock et al. [13] for flat suspended membranes (1.25–17.78 g N·m^−2^·d^−1^) and by Rothrock et al. [3] for tubular membranes (1.29–16.52 g N·m^−2^·d^−1^) without agitation and aeration of the N emitting source. The values obtained with suspended systems arranged in an annex chamber without or with slurry aeration and agitation in the treatment chamber (S3 and S8) were 12.7 and 11.4 g N·m^−2^·d^−1^, also within the range of values found for suspended systems.

Finally, as for GPM systems combining two membrane systems (S4–S9 and S5–S10), the values obtained in the present study, of 10.3 and 11.1 g N·m^−2^·d^−1^ in S4 and S9, and 10.0 and 10.6 g N·m^−2^·d^−1^ in S5 and S10, were notably higher than those obtained by Majd and Mukhtar [9], who obtained 0.9 g N·m^−2^·d^−1^ for a GPM system similar to S4. The difference can be attributed to the fact that the concentration of TAN in the slurry used by these authors was very low compared to the initial concentration of the slurry used in these trials, as explained above.

### 3.4. Economic Cost Analysis of the Different GPM Systems

Based on the goal removal and recovery rates, between 2.5 and 2.8 kg of TAN·d^−1^ (i.e., 911.3 and 1009.6 kg of TAN·year^−1^, respectively) should be recovered. With TAN recovery rates ranging from 9.5 to 20.3 g TAN·m^−2^·d^−1^, membrane surfaces between 136.1 and 282.6 m^2^ would be required to achieve this recovery.

The cost of the membrane to start up the different systems at the pilot scale would range from 15,690 € for S6 to 32,573 € for S2. The additional components, taking indicative prices reported for pilot-scale prototypes for the capture of NH_3_ from the air and the recovery of TAN from liquid media [35,36], would represent annualized costs of 18,792 € for S1, 18,792 € for S2, 30,154 € for S3, 27,654 € for S4, 30,154 € for S5, 23,007 € for S6, 23,007 € for S7, 24,994 € for S8, 31,869 € for S9, and 34,369 € for S10 (Table 2). In addition, the supplementary 10% expenses considered for the annual replacement of the membrane would range from 1569 € for S6 to 3257 € for S2.

The annual cost associated with the consumption of H_2_SO_4_ in the capture solution would range from 129.5 to 143.5 € (taking a guide price of 0.29 €·kg^−1^ [12]).

The approximate total energy consumption of the equipment used was 14.7 kWh·d^−1^ for S1 and S2; 30.4 kWh·d^−1^ for S3, S4, and S5; 59.7 kWh·d^−1^ for S6 and S7; 62.2 kWh·d^−1^ for S8; and 75.4 kWh·d^−1^ for S9 and S10. This represents annual electricity costs in the 1610 € (S1 and S2) to 8256 € (S9 and S10) range, assuming an average unit cost in Spain of 0.3 kWh·d^−1^.

Adding up all the aforementioned expenses, the estimated annual expenses for a pilot plant using gas-permeable membranes in a 1300-head pig farm would be in the 10,950 € (S1) to 20,301 € (S9) interval.

Concerning revenues, the ammonium sulfate potentially recovered per year in each system would have a value (as fertilizer) ranging from 948 € (S5, S8, and S10) to 1050 € (S9), assuming a value of 1.0 €·kg^−1^ of N as ammonium sulfate.

Therefore, the estimated net cost of ammonia recovery per year would be in the 9901 € (S1) to 19,320 € (S10) range.

A summary of investment costs, operating costs, and revenues is shown in Table 2.

No costs have been found in the literature concerning the operation of prototypes combining systems for NH_3_ capture from the air with systems for N reduction in manure, either at the laboratory or pilot scale, so this would be the first study in which such economic cost data are provided for a technology combining both systems.

The annual cost values per animal place (€·place^−1^·year^−1^) obtained in this study are of the order of those obtained with technologies such as biotrickling filters (13.2 €·place^−1^·year^−1^) or acid scrubbers (13.7 €·place^−1^·year^−1^) used to clean the air in pig housing [5]. Likewise, the values would also fall within the range of costs reported for acid scrubbers (up to 26 €/kg NH_3_ treated) [37].

The energy consumption per kg of recovered nitrogen obtained in the present study ranged from 1.86 kWh/kg N (S1) to 10.62 kWh/kg N (S10). Other technologies employed for the recovery of nitrogen from the slurry, such as stripping technology, have an associated energy consumption between 3.1 and 8.65 kWh/kg N recovered [38,39]. Therefore, systems with configurations S1 and S2 would be more efficient, configurations S9 and S10 would be more energy-intensive, and the rest of the configurations would be comparable to stripping technologies.

### 3.5. Practical Implications of the Study

In an ammonia capture system using GPMs, the membrane is the most important and expensive component of the whole system. The amount of membrane required per m^3^ of slurry to be treated and its arrangement are key elements in terms of achieving economically viable ammonia capture systems. For this reason, different membrane configurations have been investigated in this work to obtain high ammonia captures at the lowest possible cost. The reported results have an important practical implication, given that this study has been carried out in the framework of the LIFE Green Ammonia project, in which ammonia capture systems with a technology readiness level (TRL) 9 are going to be designed and built, with a view to the placing on the market of these systems.

## 4. Conclusions

Comparison of the five proposed GPM installation configurations, with and without slurry agitation and aeration, under comparable laboratory-scale working conditions, suggests that: (1) agitation and aeration, which result in the release of more NH_3_ from the slurry to the environment, leads to statistically significant differences in the mass of NH_3_-N recovered and N flux rate in the case of systems based on a single submerged GPM, a single suspended GPM, or a single suspended GPM installed in an annex compartment, but not in systems with double membranes, in which NH_3_-N removal efficiencies of ≈66.5% can be achieved without incurring extra energy costs; (2) the NH_3_-N recovery efficiency depends mainly on the chosen configuration, being higher those achieved by single submerged GPM systems and by double membrane systems (between 79 and 91%); (3) the higher capture efficiency of the more complex systems (with double membranes and/or with slurry agitation and aeration) does not compensate the associated investment and/or operating costs, respectively. Thus, with a view to the application of GPM technology at a farm scale, the simplest systems, based on single submerged or suspended GPMs installed in the same chamber where the slurry is located, would be the most advisable, presenting lower or comparable costs per kg of N recovered than those of biotrickling filters and acid scrubbers, and an energy consumption per kg of N recovered which is significantly lower than that of stripping systems.

## Figures and Tables

**Figure 1 membranes-12-01104-f001:**
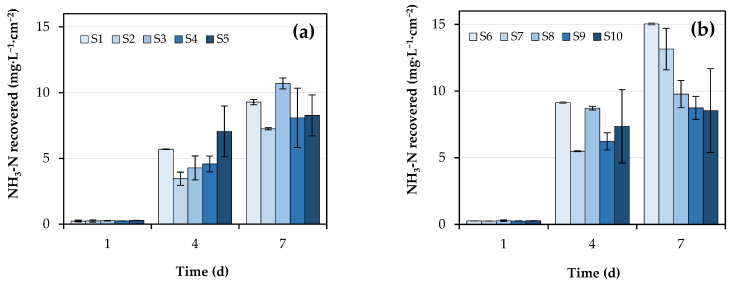
Evolution of NH_3_-N concentration in the acid capture solution per membrane surface area in (**a**) S1–S5 systems without sludge aeration and agitation; and (**b**) S6–S10 systems with slurry agitation and aeration. The meaning of the abbreviations is presented in Section 2.2.

**Table 1 membranes-12-01104-t001:** Mass balance of NH_3_-N recovered using the different configurations of the GPM system. Mass of NH_3_-N in the slurry at the beginning of the experiment (Initial NH_3_-N mass), mass of NH_3_-N emitted (NH_3_-N removed), mass of NH_3_-N recovered in the acidic solution (NH_3_-N recovered), NH_3_ removal and NH_3_ recovery efficiencies (NH_3_-N removal and NH_3_-N recovered, respectively), and N flux rate (N flux).

Conditions	GPM System	Initial NH_3_-N Mass (mg)	NH_3_-N Removed (mg)	NH_3_-NRecovered (mg)	NH_3_-N Removal (%)	NH_3_-N Recovered (%)	N Flux (g·m^−2^·d^−1^)
Without agitation + aeration	S1	4125 ± 227	1599 ± 136 b	1414 ± 21 cd	39	88	12.4 ± 0.2 cd
	S2	4308 ± 47	1567 ± 96 b	1083 ± 8 e	36	69	9.5 ± 1.7 e
	S3	4340 ± 16	2680 ± 119 a	1453 ± 111 c	62	54	12.7 ± 1.0 c
	S4	4125 ± 227	2710 ± 172 a	2361 ± 61 a	66	87	10.3 ± 0.3 cde
	S5	4006 ± 116	2692 ± 241 a	2278 ± 41 a	67	85	10.0 ± 0.2 de
With agitation + aeration	S6	4125 ± 227	2545 ± 66 a	2328 ± 10 a	62	91	20.4 ± 0.0 a
	S7	4122 ± 46	2696± 142 a	1918 ± 197 b	65	71	16.8 ± 0.1 b
	S8	4006 ± 116	2901 ± 327 a	1272 ± 20 de	72	44	11.4 ± 0.2 cde
	S9	4437 ± 20	3211 ± 29 a	2545 ± 259 a	72	79	11.1 ± 0.2 cde
	S10	4006 ± 116	2824 ± 254 a	2434 ± 2 a	70	86	10.6 ± 0.0 de

Means (n = 3) followed by a common lowercase letter are not significantly different according to Tukey’s test (*p* < 0.05). Pairs of homologous systems are highlighted in the same colors. The meaning of the abbreviations is presented in Section 2.2.

**Table 2 membranes-12-01104-t002:** Summary of investment and operating costs and revenues for each gas-permeable membrane configuration.

	S1	S2	S3	S4	S5	S6	S7	S8	S9	S10
Membrane cost (€/m)	1.88									
Membrane cost (€/m^2^)	115.24									
**Investment**										
Membrane initial cost (€)	25,686	32,573	24,547	30,923	28,774	15,690	17,625	25,240	28,719	27,145
Total initial investment (including membranes and other equipment) (€)	44,478	51,365	54,701	58,577	58,928	38,697	40,632	50,234	60,588	61,514
Annualized costs 8% interest, 10-year life (€/year)	6629	7655	8152	8730	8782	5767	6055	7486	9029	9167
**Operating Costs**										
Membrane replacement 10% (€)	2569	3257	2455	3092	2877	1569	1763	2524	2872	2715
Chemicals (€)	143	139	140	143	130	143	133	130	144	130
Power (€)	1610	1610	3329	3329	3329	6537	6537	6811	8256	8256
Total annualized costs (€/year)	10,950	12,661	14,076	15,294	15,118	14,017	14,488	16,951	20,301	20,268
Revenue for the sale of the fertilizer product (€/year)	1049	1019	1027	1049	948	1049	975	948	1050	948
**Net annual cost (€/year)**	9901	11,642	13,049	14,245	14,170	12,967	13,513	16,003	19,251	19,320
Cost per N recovered (€/kg N)	9.8	11.9	13.2	14.1	15.5	12.9	14.4	17.6	19.1	21.2
Cost per treated slurry (€/m^3^)	3.5	4.2	4.7	5.1	5.1	4.6	4.8	5.7	6.9	6.9
**Net cost (€/(place·year))**	7.6	9.0	10.0	11.0	10.9	10.0	10.4	12.3	14.8	14.9

## Data Availability

The data presented in this study are available on request from the corresponding author. The data are not publicly available due to their relevance as part of an ongoing Ph.D. thesis.

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
