# Peer review of "Comparison of the Ammonia Trapping Performance of Different Gas-Permeable Tubular Membrane System Configurations"

_membranes, 2022, doi:10.3390/membranes12111104_

Round 1
Reviewer 1 Report
Article “Comparison of the Ammonia Trapping Performance of Different Gas‐Permeable Tubular Membrane System Configurations” corresponds to the scope of the journal, touches on the topical issue of ammonia separation from wastewater, can be adopted after minor changes.
What was the reason for the flow of 0.24 L of air \ to L slurry min? Why didn't they run with different ones to find the optimal value? How did aeration go, did you achieve uniformity throughout the volume?
Based on what was the area / length of the immersed fiber in the separation phase chosen?
Why were only this type of fiber chosen?
How was the temperature control of the studied phase and the analysis system carried out? What is the error of the results obtained with this account?
According to the rules for processing experimental data, using theories of probability, you can never connect points with each other simply by lines, because. there is no information what is between them, only data approximation and other mathematical transformations are allowed. (Fig.1)
I would also like to see a more detailed screening of membranes for the components under study (ammonia, air (nitrogen, oxygen), gas humidity, (10.1016/j.proeng.2012.08.338, 10.3390/polym14112214, 10.1111/j.1749-6632.1968.tb20277.x, 10.1016/j.memsci.2009.03.013, 10.1016/j.memsci.2013.03.048, 10.1016/j.cej.2020.127726 et al., about polymeric\inorganic membranes for ammonia separation)
Author Response
REVIEWER #1
Article “Comparison of the Ammonia Trapping Performance of Different Gas‐Permeable Tubular Membrane System Configurations” corresponds to the scope of the journal, touches on the topical issue of ammonia separation from wastewater, can be adopted after minor changes.
Q1. What was the reason for the flow of 0.24 L of air \ to L slurry min? Why didn't they run with different ones to find the optimal value? How did aeration go, did you achieve uniformity throughout the volume?
Response: The aeration rate used to conduct these experiments had been successfully tested by other authors in previous studies with the same type of membrane used in these trials (e.g., 10.1016/j.jenvman.2015.01.013, 10.3390/environments6030032), finding that it provided sufficient air to effectively increase the pH of the manure while helping to prevent nitrification of
NH4+. In our study, in the aerated treatments, the air was supplied using an aquarium air pump from the bottom of the experimental chamber through a porous stone, and the supply was monitored using an airflow meter (Instruments Direct, Canton, Georgia, USA). Given that aeration was combined with agitation, uniformity throughout the volume was assured. A brief clarification on this point has been included in lines 126-130.
Q2. Based on what was the area / length of the immersed fiber in the separation phase chosen?
Response: In previous work by the same authors (10.3390/membranes11070538), favorable ammonia capture performance results were obtained using a membrane surface area of 163.4 cm2 without observing pore wetting problems, so it was decided to use the same membrane size in these experiments. This point has now been clarified in the revised version of the manuscript (line 139).
Q3. Why were only this type of fiber chosen?
Response: While we agree that the effect of different GPM technologies would be interesting to know, this study was designed to specifically focus on the evaluation of the response of different configurations of e-PTFE gas permeable membranes on the diffusion flux and NH3 removal efficiency to select the most advantageous system for industrial-scale applications, not on a comparison of different membrane technologies, and we would like to keep the focus intact. The choice of e-PTFE gas permeable membrane technology is supported by the fact that e-PTFE is a microporous, flexible, and hydrophobic material with a high permeability rate for low-pressure gas flow differentials between the inside and outside of the tube. It has demonstrated high yields in NH3 recovery from different types of sources such as chicken manure, pig manure, and anaerobically digested slurry (e.g., 10.1016/j.jclepro.2018.04.138,
10.2175/193864711802867405, 10.1016/j.wasman.2015.12.011), effectively reducing the TAN concentration of the sources. Further, it has also demonstrated high NH3 recovery efficiencies from livestock housing air (e.g., 10.13031/2013.32591, 10.1016/j.wasman.2013.03.011, 10.3390/membranes11110859). A brief clarification on this point has been included in the revised introduction (lines 49-55).
Q4. How was the temperature control of the studied phase and the analysis system carried out? What is the error of the results obtained with this account?
Response: All experiments were carried out at the same time in the same laboratory, in controlled conditions (avg. room temperature 25 °C). The temperature of the waste (slurry) and uptake (acid) solutions was monitored using a pH and temperature electrode, giving mean values of 23.4 ± 2.5 and 24.4 ± 2.4 °C, respectively. Hence, the error associated with temperature differences is expected to be negligible. This information has been included in the manuscript (lines 116-119).
Q5. According to the rules for processing experimental data, using theories of probability, you can never connect points with each other simply by lines, because. there is no information what is between them, only data approximation and other mathematical transformations are allowed (Fig.1)
Response: Thank you for pointing this out. Fig. 1 has been replotted as a vertical bar chart (as well as Figure S11).
Q6. I would also like to see a more detailed screening of membranes for the components under study (ammonia, air (nitrogen, oxygen), gas humidity, (10.1016/j.proeng.2012.08.338, 10.3390/polym14112214, 10.1111/j.1749-6632.1968.tb20277.x, 10.1016/j.memsci.2009.03.013, 10.1016/j.memsci.2013.03.048, 10.1016/j.cej.2020.127726 et al., about polymeric\inorganic membranes for ammonia separation)
Response: We thank the Reviewer for his/her kind suggestions, which we have carefully checked. We have tried at length to find the relevance of the recommended articles, but, unfortunately, they do not closely match the scope of this study, and we thus prefer not to include them.

Reviewer 2 Report
It is necessary and interesting to recovery ammonia from the waste treatment process. The authors designed 10 configuration membrane process for ammonia recovery from livestock manure. The separation performance as well as the economic feasibility was evaluated. In principle, I agree to accept the manuscript after addressing the following questions.
(1) Please indicate the meaning of “a, b, c, d, e” in Table 1.
(2) Line 145~151, it should be more reasonable to define the N removal efficiency by dividing the initial amount in the slurry by the removed mass rather than the definition of “by dividing the recovered mass by the removed mass”. Of course, the recovery and removal rate in Table 1 should be recalculated accordingly.
(3) Two-membrane system was used in S4, S5, S9 and S10. It is more reasonable to compare the NH3-N removal or recovery based on specific membrane area. For example, the unit in Figure 1 should be mg L-1 m-2 rather than mg L-1.
(4) Following last comment, the driving force for NH3 absorb is different between the one-membrane system and two-membrane system. However, there is no related information mentioned in the manuscript.
(5) Line 194: it seems wrong that the alkalinity reduced with the increase of pH of the slurry. Also the equation (2) seems weird. Please show your explanation in the main text.
(6) The "AF" misspelling on line 217 on page 5 should be "FA".
(7) How about the effect of membrane configuration on the anti-foiling performance?
(8) In principle, the chemical of H2SO4 is used to absorb NH3 for all the configuration. I am curious why there is no chemical cost for S1, S2, S4, S6 and S9 in Table 2.
Author Response
REVIEWER #2
It is necessary and interesting to recovery ammonia from the waste treatment process. The authors designed 10 configuration membrane process for ammonia recovery from livestock manure. The separation performance as well as the economic feasibility was evaluated. In principle, I agree to accept the manuscript after addressing the following questions.
Q1. Please indicate the meaning of “a, b, c, d, e” in Table 1.
Response: We have slightly rephrased Table 1 footnote, which originally indicated that “Values with different letters are significantly different at p ≤ 0.05 according to Tukey's HSD test. All values are expressed as means of n = 3. […]”. It now reads: “Means (n=3) followed by a common lowercase letter are not significantly different according to Tukey's test (P < 0.05)” (line 201).
Q2. Line 145~151, it should be more reasonable to define the N removal efficiency by dividing the initial amount in the slurry by the removed mass rather than the definition of “by dividing the recovered mass by the removed mass”. Of course, the recovery and removal rate in Table 1 should be recalculated accordingly.
Response: We appreciate the Reviewer's comment, which has brought to our attention a mistake made in the translation. In the indicated paragraph, where it says "The N removal efficiency (%) was estimated by dividing the recovered mass by the removed mass" it should actually say "N recovery efficiency" (line 162). For the calculation of "N removal efficiency", we have indeed considered the mass of NH3-N removed versus the initial amount of NH3-N present in the slurry, as indicated by the Reviewer. Therefore, the results obtained in Table 1 remain valid.
Q3. Two-membrane system was used in S4, S5, S9 and S10. It is more reasonable to compare the NH3-N removal or recovery based on specific membrane area. For example, the unit in Figure 1 should be mg L-1 m-2 rather than mg L-1.
Response: We appreciate the Reviewer’s comment. We have corrected the calculations according to the membrane surface area used in each case, and we have modified the units in Figure 1. The paragraph above Fig. 1 has been updated accordingly.
Q4. Following last comment, the driving force for NH3 absorb is different between the onemembrane system and two-membrane system. However, there is no related information mentioned in the manuscript.
Response: We do not share the Reviewer’s view on this matter. The mechanism of gas-permeable membrane technology involves the passage of NH3 (g) through a microporous, hydrophobic membrane and its capture and concentration in a generally acidic extraction solution on the other side of the membrane. Therefore, the process of NH3 diffusion between both sides of the
membrane is common to any of the systems used in the study, regardless of whether they have one or two membranes. In membranes immersed in liquid, NH3 is removed from the liquid before it is released into the air. In contrast, membranes that remain suspended in the air chamber capture NH3 (g) directly from the air. As noted in section 3.2, the difference lies in the concentration of
NH3 that is present in the gas or liquid phase, since, depending on this factor, the membrane will be able to capture more or less NH3 (lines 268-274).
Q5. Line 194: it seems wrong that the alkalinity reduced with the increase of pH of the slurry. Also the equation (2) seems weird. Please show your explanation in the main text.
Response: This explanation has been expanded in the manuscript as follows: “The recovery of NH3 (g) through the gas-permeable membrane causes an increase of acidity in the slurry as the H+ does not permeate the hydrophobic membrane. Therefore, the recovery of the NH3 by the membrane, from a net perspective, causes an acidification of the slurry and a reduction of the alkalinity. Daguerre et al., 2018, studied this behavior in further detail, concluding that the
release of OH- from the natural carbonates increased the slurry pH and promoted gaseous ammonia formation and membrane uptake; and, at the same time, the recovery of gaseous ammonia through the membrane acidified the slurry. Similar behavior was observed by other authors such as Oliveira et al (2018)” (lines 213-220).
Q6. The "AF" misspelling on line 217 on page 5 should be "FA".
Response: The typographic error has been corrected. The sentence now reads: “[…] an increase in FA concentration […]” (line 240)
Q7. How about the effect of membrane configuration on the anti-foiling performance?
Response: Membrane fouling is an important consideration that determines the lifetime of gaspermeable membranes and affects their economic viability [10.1016/j.watres.2014.02.037]. In the present work, given that experiments were only run over a week, no reduction in TAN recovery rate was detected over time, suggesting that membrane fouling (in cases where the membrane was
submerged) did not block the pores and did not affect TAN recovery. However, we have now acknowledged this important issue and suggested it as a topic for further research in the Discussion section of the revised manuscript (lines 333-338).
Q8. In principle, the chemical of H2SO4 is used to absorb NH3 for all the configurations. I am curious why there is no chemical cost for S1, S2, S4, S6 and S9 in Table 2.
Response: We thank the Reviewer for bringing this mistake to our attention. Apparently, some of the chemicals-associated costs were left black when the manuscript was translated into English. The problem has been solved in the revised version of Table 2. All values in the table have been double-checked.
